

# Lean thinking by integrating with discrete event simulation and design of experiments: an emergency department expansion

Gustavo Teodoro Gabriel, Afonso Teberga Campos, Aline de Lima Magacho, Lucas Cavallieri Segismondi, Flávio Fraga Vilela, José Antonio de Queiroz and José Arnaldo Barra Montevechi

Industrial Engineering and Management Institute, Federal University of Itajubá, Itajubá, Minas Gerais, Brazil

Corresponding author
Gustavo Teodoro Gabriel,
gustavo.teodoro.gabriel@gmail.com

## ABSTRACT

**Background**. Many management tools, such as Discrete Event Simulation (DES) and Lean Healthcare, are efficient to support and assist health care quality. In this sense, the study aims at using Lean Thinking (LT) principles combined with DES to plan a Canadian emergency department (ED) expansion and at meeting the demand that comes from small care centers closed. The project's purpose is reducing the patients' Length of Stay (LOS) in the ED. Additionally, they must be assisted as soon as possible after the triage process. Furthermore, the study aims at determining the ideal number of beds in the Short Stay Unit (SSU). The patients must not wait more than 180 min to be transferred.

**Methods**. For this purpose, the hospital decision-makers have suggested planning the expansion, and it was carried out by the simulation and modeling method. The emergency department was simulated by the software FlexSim Healthcare®, and, with the Design of Experiments (DoE), the optimal number of beds, seats, and resources for each shift was determined. Data collection and modeling were executed based on historical data (patients' arrival) and from some databases that are in use by the hospital, from April 1st, 2017 to March 31st, 2018. The experiments were carried out by running 30 replicates for each scenario.

**Results**. The results show that the emergency department cannot meet expected demand in the initial planning scenario. Only 17.2% of the patients were completed treated, and LOS was 2213.7 (average), with a confidence interval of (2131.8–2295.6) min. However, after changing decision variables and applying LT techniques, the treated patients' number increased to 95.7% (approximately 600%). Average LOS decreased to 461.2, with a confidence interval of (453.7–468.7) min, about 79.0%. The time to be attended after the triage decrease from 404.3 min to 20.8 (19.8–21.8) min, around 95.0%, while the time to be transferred from bed to the SSU decreased by 60.0%. Moreover, the ED reduced human resources downtime, according to Lean Thinking principles.

# INTRODUCTION

Improvements in the health services quality sector are significant because it directly affects the patient's satisfaction and safety. In fact, the health industry is one of the largest in the world (*Bhat, Gijo & Jnanesh, 2014*). According to the *World Bank (2019)*, in 2016, the world spending on health was around 10.0% of total GDP. The United States was the country who invested the most per capita, totaling US$ 9,869.74, followed by Switzerland. Canada is among the top 20 countries who invest in health per capita, US$ 4,458.21 in total, against the world average of 1,025.29.

Given the importance of the world health system, decision-makers seek out solutions to make more efficient and agile processes. Besides that, financial resources and skilled labor are becoming scarce. Consequently, decision-makers apply management tools for analysis and process improvements in forecasting, restructuring, and reducing costs. Indeed, many real problems are really complex. Rarely is one single approach or tool enough to reach solutions (*Brailsford et al., 2018*).

Discrete Event Simulation (DES) is one of the tools used to improve it and forecast process behavior. DES is the imitation of a dynamic process, using a computer model for evaluating, measuring and improving the performance of any system (*Harrell, Ghosh & Bowden, 2012*) without any physical risks and additional costs (*Banks et al., 2010*; *Montevechi et al., 2007*). Moreover, the literature defines DES as the model development process (hypothetical or real) aiming at performing experiments (*Negahban & Yilmaz, 2014*). Hence, predicting the behavior of real and complex systems becomes a really tough challenge, because they are influenced by a set of internal and external factors, and the experience is often unfeasible to perform (*Budgaga et al., 2016*). Therefore, DES is the first step in evaluating a change proposal, obtaining insight into a set of potential impact, and supporting management to make decisions and implement real-world improvements (*Bem-Tovim et al., 2016*; *Dengiz & Belgin, 2014*).

In addition to DES, Lean Manufacturing is also efficient in improving processes. Lean emerged in the 1950s when the Japanese automotive industry faced a challenging scenario. The market was limited and a wide variety of vehicles were demanded. It was in contrast to the current philosophy of mass production until then (few varieties produced on a large scale). Furthermore, Japan's economy was weakened, with low capital availability and few international trade relations. Also, the acquisition of modern western production technologies became harder (*Womack, Jones & Roos, 1990*). To meet this challenge, the Japanese automaker company Toyota began developing a new system of production, the Toyota Production System (TPS). Taiichi Ohno, credited as the facilitator of the TPS, aimed to serve customers in the shortest time, at the highest quality and the lowest possible costs. Thereby, it would be necessary to focus all effort on activities that add value to the customer and eliminating wastes (*Graban, 2016*). Over time, Lean has expanded throughout other sectors, such as healthcare, which has been renamed to Lean Healthcare (LH).

Finding wastes in manufacturing has been adapted to health services. Defects correspond to poor medication administration or incorrect doses, while overproduction characterizes as unnecessary diagnostic procedures. Inefficient transportation is inadequate layouts and

laboratories away from collection points. The inappropriate layout may also link to the unnecessary movement of nurses and doctors. Waiting features idle employees with uneven workloads and patients waiting for service. The stock is characterized as expired supplies and super-processing, e.g., data in the patient registry that will not be used later (*Graban, 2016*).

Finally, performing experiments with DES is essential for the analysis of the current scenario and for proposing improvements. The experiment is indicated when the modeler desired to optimize the process (*Banks et al., 2010*). A good experiment demand to set the parameters so the responses may approach the required values with the lowest variability (*Montgomery & Runger, 2018*). The best way to evaluate several factors in a process is by using suitable techniques to plan the experiments, e.g., factorial experiments, Taguchi, and Plackett-Burman (*Montgomery & Runger, 2018*). In this case, the factors change simultaneously, observing if there is an interaction between them. Moreover, it generally requires fewer tests than the "best guess" strategy, where the expert performs random experiments. According to *Banks et al. (2010)*, with some adaptations in the factorial arrangements, we may evaluate the individual, interactions, and quadratic effects.

Thereby, the three presented techniques, when used together, show effective results in health processes. In this sense, this study aims to plan the expansion of a Canadian emergency department (ED). The expansion is necessary because four nearer small centers were closed and all the patients needed to be attended to in the new ED. According to LH principles, the study aims to define the ideal number of resources, beds for care, and beds in the Short Stay Unit (SSU). The project purposes of reducing the patients' Length of Stay (LOS) in the ED. Additionally, they must be taken care of as soon as possible after going through the triage. Furthermore, it is necessary to determine the ideal number of SSU beds because the patients should not wait for more than 180 min for their transfer.

In order to contribute to the literature, the study uses the Design of Experiments (DoE) to determine the resource numbers influence on each shift, in addition to identifying the main limiting factors of physical resources for the ED expansion. The simulation also allows evaluating the influence of demand variation throughout the day and week in the future scenario.

The article is divided as follows: the next section presents a literature review of DES and Lean in hospital environments, followed by the methodology. The subsequent sections present the results and discussions to conclude the study.

## LITERATURE BACKGROUND

This section describes the studies that used Discrete Event Simulation and Lean Healthcare to improve the healthcare sector. Moreover, this section presents the two main tools used in our paper, the IDEF-SIM technique and the Design of Experiments.

### Discrete Event Simulation and Lean in healthcare systems

Performing DES in healthcare is not new, dating back to the 1960s (*Pitt, 2008*). However, since then, there has been considerable growth for its interest. According to *Arisha & Rashwan (2017)*, this progress is strong evidence that the simulation provides better

decisions in health services management without compromising patient safety. This advantage has increasingly attracted the attention of hospitals and health authorities (*Cheng et al., 2017*). However, experts in healthcare simulation claim that its use is more complicated than in other areas (*Tako & Robinson, 2015*). The main problems found in the healthcare simulation are less evident structure; the system is complex; more significant effort to collect and access data; barriers due to ethical issues; client's shortage of time; and more difficulty in ensuring implementation.

Despite the difficulties, it is possible to find studies with positive results linked to costs, capacity, wait and stay time, and levels of service and losses. *Zhou & Olsen (2018)* applied DES for medical supplies management and decreased the costs involved in the process by reducing expired drugs. *Hussein et al. (2017)* presented effective results reducing overcrowding in a hospital. Similar results were shown by *Babashov et al. (2017)* and *Shim & Kumar (2010)* by decreasing patients' waiting time in an emergency department. *Rau et al. (2013)* and *Uriarte et al. (2017)* also used DES to reduce patient waiting time in treatment centers and the radiotherapy sector, respectively. Furthermore, *Al-Araidah, Boran & Wahsheh (2012)* applied the tool in an ophthalmology laboratory to decrease patients' waiting and appointment time. Regarding planning and capacity analysis, *Pinto et al. (2015)* defined the ideal number of beds for a Brazilian hospital. In the balancing of staff work, *Reynolds et al. (2011)* applied the simulation to reduce the workload of an English pharmacy and (*Pongjetanapong et al., 2019*) used the tool to evaluate the change effect to staff levels in a cytology department.

Although these studies present positive results, we found some limitations in them. Some papers did not consider variables and statistical distributions for different workgroups (*Reynolds et al., 2011*) and processes (*Rau et al., 2013*). *Pinto et al. (2015)* state that the simulation results have a slight underestimation, although it seems reliable and more effective for decision making than empirical calculations. Although articles use powerful tools jointly with DES, some studies do not test the interaction of resources, locations, and factors using the "best guess" approach (*Al-Araidah, Boran & Wahsheh, 2012*; *Uriarte et al., 2017*). In contrast, *Hussein et al. (2017)* use DoE to observe the interaction of two factors in their scenarios. Nevertheless, they did not observe the interaction in the resource number in each shift (present in this study). *Shim & Kumar (2010)* state that the simulation did not take into account the varying resource capacities (doctors and nurses) and locations (workstations from the sorting station to the payment station) involved in the emergency care process. Finally, *Babashov et al. (2017)* use simulation to determine the average waiting time for all patients overall, not for each patient type.

It was also possible to verify improvements using only LH. Even when not implemented systematically and comprehensively in the organization, LH can provide several benefits to health services (*D'Andreamatteo et al., 2015*). Among these benefits, the authors highlight improvements in productivity, costs, financial results, quality of service delivery, and patient and team safety and satisfaction. However, implementing Lean is long-lasting and challenging work in the health sector (*Toussaint & Berry, 2013*). Lean transform the organizational culture from the inside out, requiring managers and leaders to become

facilitators, mentors, teachers, and enable employees to take the initiative in making improvements.

Studies that show LH implementation can be found for the elimination of processes that do not add value to the patient (*Teichgräber & Bucourt, 2012*) and reduction in the instrument collection stage (*Kimsey, 2010*). *Laganga (2011)* used lean methodology to increase patient care capacity, while *Papadopoulos, Radnor & Merali (2011)* used LH to reduce delays in receiving samples, prioritize urgent work, standardize processes and anticipate problem identification. Although LH application is useful and essential in waste elimination and increasing patient value-added time, it can be time-consuming, and the staff involved may resist the methodology.

When used jointly, Lean and DES have three goals: to teach, evaluate, and facilitate the process. In order to evaluate, it allows the execution of experiments and the evaluation of their results. It should be employed after holding a team meeting, testing ideas, and creating new solutions (*Robinson et al., 2012*). The use of DES and LH in hospital settings may bring more quality and efficiency to patients and management (*Gaba, 2004*). Also, the patient flow may be optimized and served as a motivational factor for employees (*Salam & Khan, 2016*). *Swick et al. (2012)* state that hospitals that integrate these tools offer an efficient method of strategic planning and provide employees with a privileged view of how to reduce waste and add value. Moreover, it is possible to reduce patients' waiting time, decreasing employee workload, and promoting resource reallocation (*Haddad et al., 2016*; *Bhat, Gijo & Jnanesh, 2014*). Finally, *Doğan & Unutulmaz (2016)* and *Robinson et al. (2014)* used the tools to transform static mappings into dynamic.

Some limitations were found in these studies. *Salam & Khan (2016)* did not use arrival and staff modeling. *Swick et al. (2012)* limited the work to the daytime hours (more considerable resources demand), and there was no adjustment for the night shift. *Bhat, Gijo & Jnanesh (2014)* conducted the study in an environment where patient arrival cannot be adequately controlled. *Haddad et al. (2016)* conducted the study limited to a process that ignores interdependencies with internal and external parts and laboratory tests. *Doğan & Unutulmaz (2016)* used only Value Stream Mapping and did not carry out planned experiments.

## IDEF-SIM Technique

The IDEF-SIM (Integrated Definition Methods - Simulation) process mapping technique was developed by *Montevechi et al. (2010)*. It aims to support modeling work in the implementation and analysis phase, reducing project time. The IDEF-SIM was created based on existing logical elements in other mapping techniques, such as IDEF0, IDEF3 and flowchart (*Peixoto et al., 2017*; *Montevechi et al., 2010*). To our best knowledge, there is no history of this technique used in healthcare systems, although we have found in manufacturing (*Pereira et al., 2015*; *Montevechi et al., 2010*) and transports (*Lopes et al., 2017*). Nevertheless, we chose to use this technique because the symbols are directly translated from the conceptual model for the computer model, presenting elements of the simulation, e.g., entities, locations, resources, functions, controls, logical rules, and

transport (*Pereira et al., 2015*). The symbols used in the IDEF-SIM are (*Peixoto et al., 2017*; *Montevechi et al., 2010*):

a) Entity (circle): items processed in the system, e.g., raw materials, products, people.
b) Functions (rectangle): places where entities change or have delays, e.g., workstations, conveyors, queues, stocks, beds, waiting lines.
c) Entity flow (arrow): direction that the entities follow in the model.
d) Resources (arrow under the rectangle): elements that move the entities or perform the functions, e.g., people, such as operators, nurses, doctors.
e) Controls (arrow above the rectangle): rules used in the functions, e.g., rules of scheduling, queuing, sequencing.
f) Movement (large arrow): it is used when there is a significant entity displacement.
g) Explanatory information (dashed arrow): it is used to include some explanation.
h) Rules for parallel and/or alternate flows: entities must follow in parallel (AND –"&"); alternatively (OR –"X"); or in parallel and/or alternate paths (AND/OR –"O").
i) Input flow (initial arrow): determines the input or entities' creation.
j) End of the system (painted circle): determines the end of a path.
k) Connection with another figure (triangle): divide the model into different figures.

## Design and analysis of experiments

Design of Experiment (DoE), is considered one of the most relevant methodologies for researchers performing applicable experiments. Moreover, according to *Montgomery (2017)*, DoE is a collection of statistical techniques capable of generating and analyzing experimental projects in which several factors are varied at the same time. In DoE, the modeler chooses different levels for the independent variable and can vary the factors simultaneously to study the effects in the responses (dependent variables). These effects can be separate from each independent variable or the interaction between them (*Baril, Gascon & Vadeboncoeur, 2019*). When experimental designs are appropriately designed, trial and error analysis are avoided. Also, it is possible to detect unimportant factors and simplify the simulation model. Among DoE techniques, the $2^k$ factorial design is used as it requires few executions per factor, and data interpretation can proceed mainly through arithmetic and graphs. In DoE, *k* value is the number of factors that presented the upper (+) level and lower (−) level. In the literature, many studies use DoE to plan and dimension resources around healthcare systems and EDs. *Baril, Gascon & Vadeboncoeur (2019)* used the DoE to vary the number of doctors, nurses and beds to decrease LOS in Canada. *Bhattacharjee & Ray (2018)* used DoE and DES to gain insights into appointment scheduling. Moreover, *Gul, Guneri & Gunal (2016)* dimensioned an emergency department to support natural disasters analyzing the number of nurses and doctors. Although these articles use DoE, our study is different because we use DoE to dimension beds and the number of resources in each shift.

## MATERIALS & METHODS

Modeling and simulation, used in the study, comprises three phases: design, implementation, and analysis (*Montevechi et al., 2010*). In the first phase, i.e., design,

we should formulate the problem. In this step, we define the process to be modeled, actions and goals. The second step is the conceptual model building, validation, and documentation. It is possible to use many languages, but opting for a simulation-oriented is desirable (*Montevechi et al., 2010*). The last stage is the input data modeling, e.g., time, cost, percentages, capacities, varying according to the study (*Banks et al., 2010*; *Montevechi et al., 2010*).

The second major phase, implementation, covers the steps of computer model building, verification, and validation. The modeler must use familiar software for computer model building. Thus, it is performed the verification, ensuring that the computer model programming corresponds to the conceptual model (*Sargent, 2013*). Finally, it is carried out the validation through hypothesis tests, confidence intervals, or sensitivity analysis (*Sargent, 2013*).

For the analysis, the modeler starts the planning, construction, and analysis of the experiments. In this step, the modeler defines possible scenarios, besides the use of experiment planning and statistical tests (*Montgomery & Runger, 2018*). After the experiments, the scenarios are analyzed, and the team obtains the conclusions and answers to the problem.

Based on the three major steps described, we conduct the study as follows:

1. Conceptual Modeling: we developed the conceptual model using the IDEF-SIM technique. We choose this language because it is considered specific for DES, facilitating computer programming (*Montevechi et al., 2010*).
2. Computer Modeling: we used FlexSim Healthcare® software.
3. Model Execution and initial analysis: we performed replicas to confirm the patient and the hospital unit's performance measure in the initial planning scenario. These executions allowed us to choose the first set of DoE decision factors.
4. Design and analysis of experiments: we chose a complete factorial design because it requires few runs per factor and a better fit model. It was carried out a DoE to find out the ideal number of employees at each shift.
5. Confirmation runs: the optimal values of the decision variables found in the experiments were used in the model to confirm the results.

## RESULTS AND DISCUSSION

### Case study

Cape Breton Regional Hospital (CBRH) is one of four facilities that make up the Cape Breton Health Care Complex, a Canadian public health network of four hospitals, serving approximately 100,000 inhabitants. This paper aims to plan the expansion of a Canadian ED to meet the demand that comes from the small closed care centers. The exponential increase in demand is due to the closure of these small service units in locations near the CBRH.

Regarding the patients' flow, they arrive at the ED by themselves or by ambulance. Both are triaged and ranked according to severity. This classification is given according to the Canadian Triage Acuity Scale (CTAS). CTAS level I corresponds to the most severe level,

resuscitation (blue, to be seen immediately); level II is emergent (red, to be seen <15 min); level III is urgent (yellow, to be seen <30 min); Level IV is less urgent (green, to be seen <60 min) and level V is non-urgent patient (white, to be seen <120 min). Patients arriving by ambulance, after triage, go straight to care. Patients who arrive on their own go to the registry and expect the transfer to their appointment. CTAS level I, II, and III patients go to a bed area (45 beds and 3 for mental health), while CTAS IV and V follow to the vertical area (10 seats).

When the patient arrives at the bed area (BA) or the vertical area (VA), he undergoes two evaluations made by the nurse and the physician. Then the ward clerk receives and records the patient's prescription. These prescriptions may include laboratory tests (blood, urine), medical procedures, and diagnostic imagining (DI). Such procedures are assigned according to the patient's category, shown in Table 1. If DI is required, a nurse and a porter must escort CTAS level I patients. For other patients, only the porter is required.

After waiting for the exam results, the patient may go through an appointment with a specialist or move on to the next stage. If the patient goes through the appointment, he must wait until the specialist arrives, since they are not in the ED. In the next step, the patient may be discharged or go through monitoring. After monitoring, the patient goes to the SSU (10 available beds), where it can remain from one to three days.

The ED Nurses are divided into two teams. The first team is responsible for patients CTAS level I and II (EDN1), and the second is responsible for CTAS level III, IV, and V (EDN2). They work in four shifts: 07:00 a.m. to 07:00 p.m.; 09:00 a.m. to 09:00 p.m.; 11:00 a.m. to 11:00 p.m. and 07:00 p.m. to 07:00: a.m. Physicians and Triage Nurse (TN) start working at the same shifts. In addition, three porters work in the following shifts: 07:00 a.m. to 07:00 p.m.; 10:00 a.m. to 06:00 p.m. and 07:00 p.m. to 07:00 a.m.

The study aims to analyze whether the ED can absorb the full demand from the closures. Other issues to be solved are ideal numbers of resources (TN, EDN, physicians, and porters); the ideal number of beds in BA, chairs in VA and beds in SSU; reduction in the time of care after going through the triage, in the time to transfer from BA/VA to SSU and LOS. In this sense, the decision-makers from the hospital proposed an approach to be tested by the simulation, called Initial Planning (IP). We used DES because patients are individual entities with particular characteristics, and the procedural flow depends on the CTAS of each one. Moreover, we chose DES over System Dynamics because it has the most relevant characteristics of the work responses. We can say that DES is better in decision-making, prediction, optimization, and comparison; it behaves well for high rates of change and presents the high importance of tracking individuals. In addition, it is suitable for systems with few entities and relatively small-time scales (*Braislford & Hilton, 2001*).

## Conceptual modeling

We performed the conceptual modeling of the system through the IDEF-SIM language. Figure 1 shows the patient flow. We do not present percentages for each process because there are many combinations based on the disease category and CTAS level.

**Table 1  Patient category and specialist.**

| Treatment category | Specialist | Proportion |
| --- | --- | --- |
| General/Minor issues (GMI) | – | 17.7% |
| Respiratory (RE) | Respiratory Therapist | 14.6% |
| Gastroenterology (GA) | Gastroenterologist | 11.9% |
| Orthopedic (OR) | Orthopedist | 10.3% |
| Cardiology (CA) | Cardiologist | 9.7% |
| Dermatology (DE) | Dermatologist | 8.3% |
| Genitourinary (GE) | Urologist and Nephrologist | 6.1% |
| Ear, nose, and throat (ENT) | Otolaryngologist | 4.9% |
| Mental Health (MH) | Crisis and Psychologist | 4.3% |
| Neurologic (NE) | Neurologist | 4.2% |
| Ambulatory Return Visit (ARV) | – | 4.0% |
| Ophthalmology (OP) | Ophthalmologist | 1.8% |
| Gynecology (GY) | Gynecologist | 1.1% |
| Substance Misuse (SM) | Crisis and Psychologist | 0.9% |

As mentioned in "IDEF-SIM Technique", the first circle represents the entities in the model, i.e., patients. Rectangles mean delays in patient flow or procedure waiting (queue, exam results). Large arrows represent transport for the patient, e.g., an exam performed in another room. Processes that need a nurse or doctor are indicated with an arrow at the bottom of the rectangle, while the upper arrows are controls. Symbols with an "x" inside them indicate an option (OR connection), while symbols with an "&" represent an obligation (AND condition). The diamonds represent a stopping point. The entity only follows the flow if the given conditions have been met. In the model, the patient only continues the flow if the test results are ready, and there is an evaluation by a doctor and a nurse. We ensure that the conceptual model presents high reliability since it was validated by face-to-face validation, where experts verify if the model matches the real system (*Kleijnen, 1995*; *Sargent, 2013*).

### Data set

Data collection and modeling were based on historical data (patients' arrival) and from some databases that are used in the hospital from April 1st, 2017 to March 31st, 2018. Through the historical data, we observed that patients arrive at different rates hourly and daily. These rates are repeated weekly. As soon as a patient is screened in the emergency room their information is entered into the system. After that, Medietch® is used for registration, diagnostic codes, CTAS level, treatment time, diagnostic test ordering, and blood exams. Figure 2 shows the patient's arrival rate based on weekday and the two-hour scale.

We notice that the curves show similarity in the shape where the number of patients starts to increase from 08:00 a.m. Moreover, the curves present a peak between 10:00 a.m and 12:00 p.m. The days with the highest arrival rates are Tuesday, Wednesday and Thursday. Regarding the type of patient, the curves do not differ significantly, showing

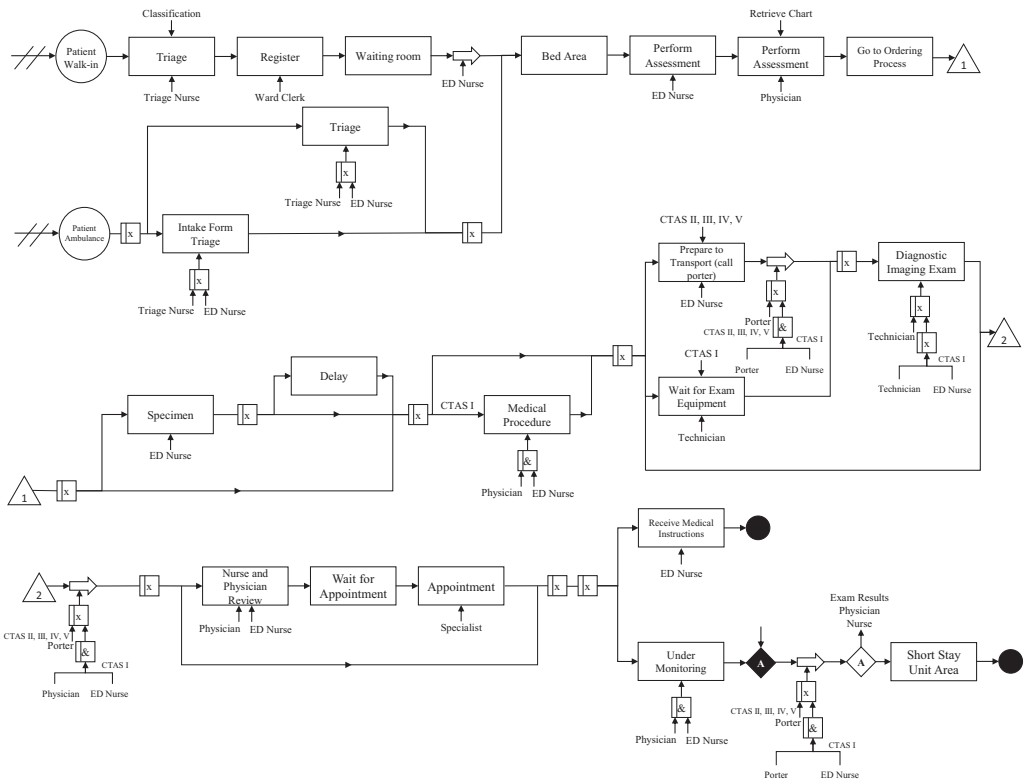

**Figure 1  Conceptual modeling.**

the same behavior throughout the day and week. Patients of CTAS level III and IV are predominant in the ED. As shown in Fig. 1, the patient may be admitted for SSU placement or be discharged. Each of these stages involves different resources, such as doctors, nurses, porters, and the patient suffers a delay (process time). These times were estimates from the ED manager and nurses through Delphi method (Table 2).

As previously stated, ED patients may require a sample, consultation and/or lab work. The patients may carry out a sample, e.g., blood work (BW) and specimen (SP). In addition, patients may have DI exams, such as X-ray (XR), electrocardiogram (EKG), echocardiogram (ECHO), computerized tomography (CT), and ultrasonography (US) according to the CTAS level and patient's category (Table 1). Table 3 summarizes the probability if the patient needs an exam.

There is a probability of the patient having an appointment with one or two specialists. Moreover, when the patient needs an appointment with an expert, some response delay can happen according to the specialist and the CTAS patient level. This delay occurs because the specialist is not staffed in the ED unit all the time. They release the patient to the attending physician while consulting the patient unless they are CTAS level I. After a specialist arrives in the ED to consult with a patient, he attends all patients that need him, from lowest to highest CTAS level. The specialist will start a new consult any time before the end of their shift. Any patients still in need of a consult at the end of the day will be seen by Internal

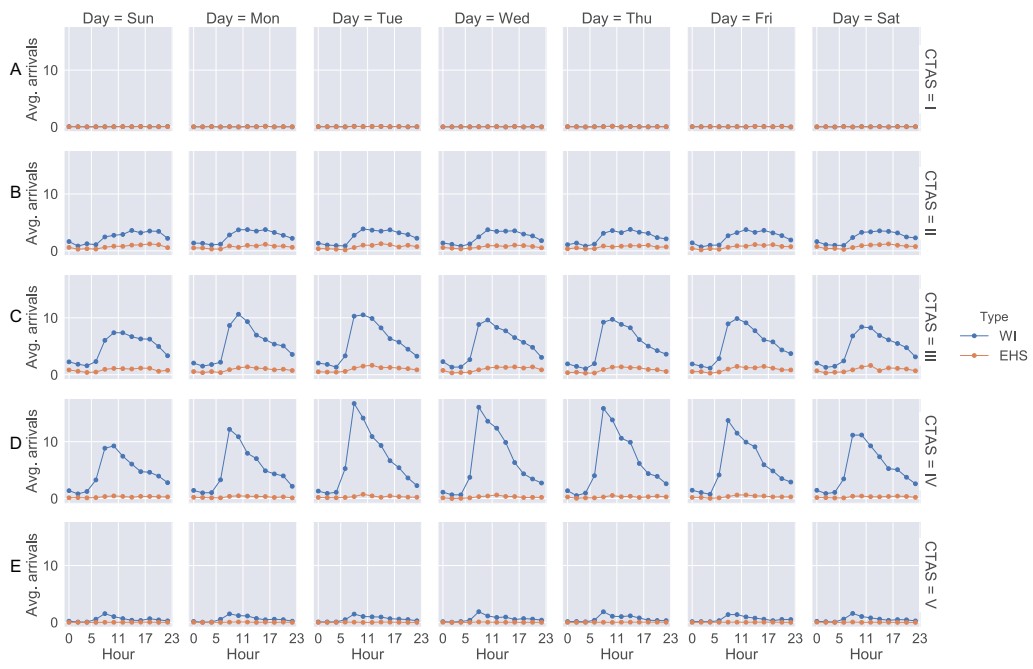

**Figure 2** **Arrival rate for each patient CTAS level.** (A) CTAS level I patient arrivals during the week. (B) CTAS level II patient arrivals during the week. (C) CTAS level III patient arrivals during the week. (D) CTAS level IV patient arrivals during the week. (E) CTAS level V patient arrivals during the week.

Medicine or wait until the beginning of a specialist's shift the following day to receive an appointment. Internal Medicine covers cardiology, nephrology and gastroenterology. Table 4 describes these data.

## Computer Modeling

We designed the computer model in FlexSim Healthcare® software, programming three different processes. First, we programmed patient flow according to CTAS levels, because each level may indicate different paths and require different resources. The second process involves the procedures exam, from sample collection to the end result. It directly impacts the patient waiting time, since it can only follow through the flow after the results, characterizing a necessary time, but that does not add value for them. Finally, we construct global processes, such as staff meetings that affect patients' waiting time and must be reduced to a sufficient level. Figure 3 shows the computer model designed for the IP scenario.

After the model was built, we performed a verification process. The process was conducted through a face-to-face technique for each CTAS level. The specialists and the modeling team analyzed the simulated model and agreed that it was as close to the real patient flow as possible. We also performed the validation. If the modeled system is a non-observable system, it is recommended using a subjective approach (*Sargent, 2013*; *Sargent, Goldsman & Yaacoub, 2016*). In this sense, we carried out a sensitivity analysis to see the model variability. We have tested patient arrival rates at five levels: 100%, 75%, 50%,

**Table 2  Processing time and required staff.**

| Task | Process time (min) | Triage nurse | Registration | ED nurse | Physician | Admin | Porter |
|---|---|---|---|---|---|---|---|
| Triage | T(4,7,10) | x | | | | | |
| Escort to Bed | T(0.5,1,2) | | | x | | | |
| Register Patient | 3 | | x | | | | |
| Nurse Assessment | CTASIV/V: 5; CTASIII: 10; CTASII: 15 | | | x | | | |
| Physician Assessment | T(7,12,15) | | | | x | | |
| Write Patient Orders | T(2,4,5) | | | | x | | |
| Initiate DI/Lab | T(2,2.5,3) | | | | | x | |
| Prepare Patient for DI | 3 | | | x | | | |
| Receive/Add Tests to Chart | 1 | | | x | | x | |
| Review Tests (Nurse) | 1 | | | x | | | |
| Review Tests (Physician) | 2 | | | | x | | |
| Take Specimen | T(4,6,15) | | | x | | | |
| Medical Procedure[a] | T(15,20,60) | | | x | x | | |
| Initiate Consult | 2 | | | | | x | |
| Arrange Consultant | 2 | | | | | x | |
| Discuss with Specialist | 3 | | | | x | | |
| Make Disposition | 20% - 30; 80% - 3 | | | | x | | |
| Admission Form | 2 | | | x | | | |
| Determine Care Plan | T(7,10,12) | | | | x | | |
| Contact Specialist | 5 | | | | x | | |
| Transport Patient to SSU (I and II)* | T(15,20,25) | | | x | | | x |
| Care Plan and Final Orders | T(2,4,10) | | | | x | | |
| Give Instruction | T(2,5,12) | | | x | | | |
| Prepare Patient | T(1,2,4) | | | x | | | |
| Shift Change Written/Verbal | 5 | | | | x | | |
| EHS Call | 1 | x | | x | | | |
| EHS Triage | T(4,7,10) | x | | x | | | |
| Prepare Room for Next Patient | 1 | | | x | | | |
| Huddle[a] | 10 | x | x | x | x | x | |

**Notes.**

[a]Denotes a task which requires all staff indicated (x).

25% and 10%. We got the mean LOS and the percentage of treated patients to perform the validation. For LOS, in minutes, the simulations show: 2,213.7 (100%), 428.2 (75%), 352.9 (50%), 270.9 (25%), 243.4 (10%). We expected to increase the number of patients, LOS would increase in the same way. This situation occurs because the patients have to wait more time for available resources. Although, if we are decreasing the arriving rate, the percentage of treated patients starts to increase because there are more staff available for each one. Figure 4 shows the sensitivity analysis for the validation.

**Table 3 Exam probability for each CTAS level.**

| Exam | CTAS | Treatment category | | | | | | | | | | | | |
|---|---|---|---|---|---|---|---|---|---|---|---|---|---|---|
| | | GMI | RE | GA | OR | CA | DE | GE | ENT | MH | NE | OP | GY | SM |
| | I and II | 1% | x | ✓ | 50% | x | x | ✓ | ✓ | x | ✓ | 50% | x | x |
| CT | III | 1% | x | 50% | 10% | x | x | 50% | x | x | x | x | x | x |
| | IV and V | 1% | x | x | x | x | x | x | x | x | x | x | x | x |
| | I and II | x | x | x | x | 10% | x | x | x | x | x | x | x | x |
| ECHO | III | x | x | x | x | x | x | x | x | x | x | x | x | x |
| | IV and V | x | x | x | x | x | x | x | x | x | x | x | x | x |
| | I and II | x | ✓ | x | 75% | ✓ | 25% | x | x | x | x | x | x | x |
| EKG | III | x | x | x | 75% | ✓ | x | x | x | x | x | x | x | x |
| | IV and V | x | x | x | x | ✓ | x | x | x | x | x | x | x | x |
| | I and II | 1% | x | ✓ | x | x | x | ✓ | x | x | x | x | ✓ | x |
| US | III | 1% | x | 50% | x | x | x | 50% | x | x | x | x | ✓ | x |
| | IV and V | 1% | x | x | x | x | x | x | x | x | x | x | x | x |
| | I and II | 50% | ✓ | ✓ | ✓ | ✓ | 25% | ✓ | ✓ | x | ✓ | 25% | x | x |
| XR | III | 50% | ✓ | 50% | ✓ | 75% | x | 50% | 25% | x | x | x | x | x |
| | IV and V | 50% | 25% | x | 50% | 75% | x | x | x | x | 25% | x | x | x |
| | I and II | 50% | ✓ | ✓ | ✓ | ✓ | 75% | ✓ | ✓ | ✓ | ✓ | ✓ | ✓ | ✓ |
| BW | III | 50% | ✓ | ✓ | ✓ | ✓ | 25% | ✓ | 25% | ✓ | ✓ | x | ✓ | ✓ |
| | IV and V | 50% | x | x | x | 25% | x | x | x | 50% | 10% | x | 50% | 75% |
| | I and II | x | ✓ | 50% | 25% | x | 25% | ✓ | 25% | ✓ | ✓ | x | ✓ | ✓ |
| SP | III | x | 75% | 50% | 10% | x | 25% | ✓ | 50% | ✓ | 50% | x | ✓ | ✓ |
| | IV and V | x | 25% | x | x | x | x | ✓ | x | 50% | x | x | 50% | 75% |

## Model execution and initial analysis

After the computer validation, the model was simulated for one week, and data were collected from Tuesday to Tuesday. A warm-up of one day (Monday) was necessary to get a stable process. Consequently, according to the computer simulation, the ED cannot meet the expected demand for IP. Moreover, the obtained results were:

- On average, 1,504 patients arrived in the model;
- On average, 258 (17.2%) patients were completed treated in the simulated period;
- Around 1,157 patients (76.9%) did not even make it triage;
- On average, LOS was 2,213.7 min;
- Patients wait about 404.3 min to be seen after triage;
- Patients who need to go to SSU wait, on average, 367.4 min to be transferred.

Regarding the results, the number of patients that do not go through the risk classification is alarming. In addition, after triage, the average waiting time is around 404.3 min, which corresponds to approximately 3.5 times what the patient with CTAS level V should wait at most. Moreover, patients expect to be attended after the risk classification, on average

**Table 4 Specialist probability appointment for each CTAS level and data set.**

| Treatment category | Consult 1 | CTAS I and II | | CTAS III | | CTAS IV and V | |
|---|---|---|---|---|---|---|---|
| | | C1 | Time (min) | C1 | Time (min) | C1 | Time (min) |
| RE | Respiratory Therapist | 75% | 60 | 10% | 15 | x | x |
| GA | Gastroenterologist | ✓ | T(15,22,30) | 25% | T(15,22,30) | x | T(15,22,30) |
| OR | Orthopedist | ✓ | 10 | ✓ | 5 | 50% | 3 |
| CA | Cardiologist | ✓ | T(30,45,60) | 50% | T(30,45,60) | 10% | T(30,45,60) |
| DE | Dermatologist | 25% | T(30,45,60) | 10% | T(30,45,60) | x | x |
| GE | Urologist | ✓ | T(15,22,30) | ✓ | T(15,22,30) | x | x |
| ENT | Otolaryngologist | ✓ | 60 | 50% | 20 | x | x |
| MH | Crisis Response | ✓ | T(60,75,90) | ✓ | T(60,75,90) | ✓ | T(60,75,90) |
| NE | Neurologist | ✓ | 60 | ✓ | 60 | x | x |
| OP | Ophthalmologist | 10% | T(30,45,60) | 5% | T(30,45,60) | x | x |
| GY | Gynecologist | ✓ | T(15,22,30) | ✓ | T(15,22,30) | x | x |
| SM | Crisis | ✓ | T(60,75,90) | ✓ | T(60,75,90) | ✓ | T(60,75,90) |

| Treatment category | Consult 2 | CTAS I and II | | CTAS III | | CTAS IV and V | |
|---|---|---|---|---|---|---|---|
| | | C2 | Time (m) | C2 | Time (m) | C2 | Time (m) |
| GE | Nephrologist | 50% | T(10,15,20) | 25% | T(30,45,60) | x | x |
| MH | Psychologist | ✓ | T(60,75,90) | 75% | T(60,75,90) | 25% | T(60,75,90) |
| SM | Psychologist | ✓ | T(60,75,90) | 50% | T(60,75,90) | 25% | T(60,75,90) |

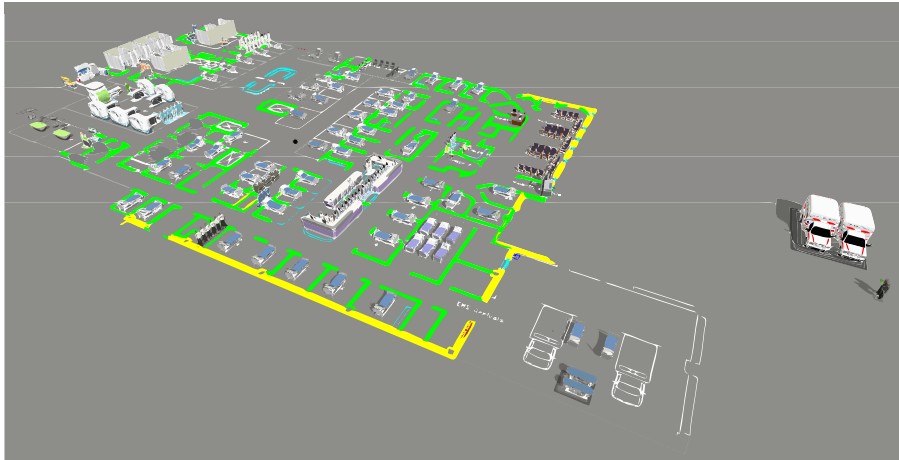

**Figure 3 Computer model screen in the IP scenario**

25.9, 164.6, 332.2, 649.7 and 419.5 min for the CTAS levels I, II, III, IV, and V, respectively, which does not meet the specifications.

The result is unsatisfactory and below the current quality standard offered by the hospital. We conclude the initial ED planning for the possible scenario after closing the surrounding care centers still has deficiencies. This reinforces the importance of using

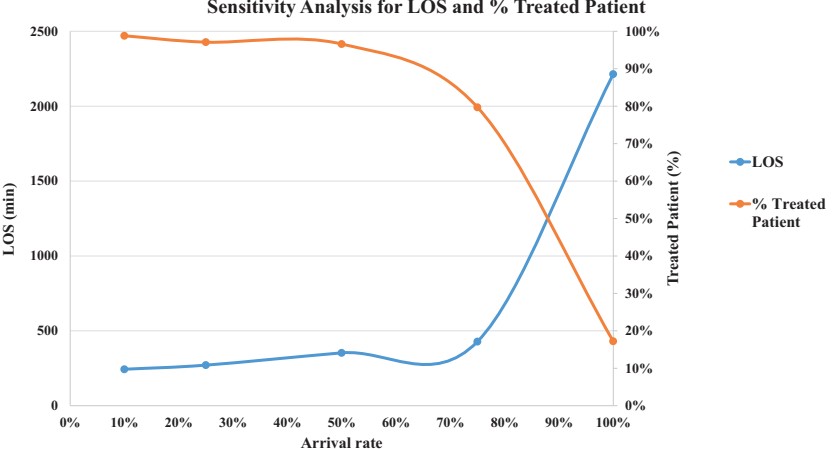

**Figure 4** **Sensitivity analysis for validation (LOS and % Treated Patient).**

simulation to ensure that ED, in this scenario, continues to offer a high quality of care. Especially in disruptive situations where planning becomes more difficult.

Moreover, for the initial analysis, most of the resources are idle, e.g., triage nurses and physicians. Most nurses and porters are waiting for empty locations (beds and seats). Consequently, the patient's flow is stuck. It other words, beds in the BA, MBA, SSU, and VA are occupied most of the time.

In this sense, based on LT, some improvements have been proposed. These improvements aim to reduce patient waiting time, thereby increasing the safety of their care. Therefore, we investigated the interaction between places in the ED and the resources in the patient flow.

## Design and analysis of experiments
### Design and analysis of experiments for locations

For the first set of experiments, we defined a complete factorial $2^4$, where the factors are the number of beds in the SSU, Mental Bed Area (MBA), Bed Area (BA), and the number of seats in the Vertical Waiting Area (VA). We estimated these numbers using Eq. (1):

$$Q_i = \frac{t_i * p_i}{60\text{min}} \tag{1}$$

Where:

$Q_i$ = Required number of location or staff per patient/hour;

$t_i$ = Time in minutes that a patient stays in the area or needs a resource (average);

$p_i$ = number of patients that arrives in area $i$ or needs a resource (average);

Thus, considering these variables, a full factorial design was planned. We choose a full factorial design ($2^4 = 16$ runs) since the number of runs is lower, and design reliability is higher than half factorial ($1/2*2^4 = 8$ runs). To determine the levels of DoE, we used a 20% margin of factor target to define "level −" and "level +". The results were rounded down and up, respectively, and are shown in Table 5.

**Table 5  Variables related to locations and DoE parameters.**

| Variable | Number of location | | DoE parameters | | |
|---|---|---|---|---|---|
| | IP | Ideal | Low level | Centre point | High level |
| **SSU** | 10 | 30 | 10 | 20 | 30 |
| **BA** | 45 | 49 | 45 | 49 | 53 |
| **MBA** | 3 | 4 | 3 | 4 | 5 |
| **VA** | 10 | 13 | 10 | 13 | 15 |

The performance measures defined in response to DoE were: (i) Weekly treated patients; (ii) LOS; (iii) Time for the patient to be attended after triage and (iv) Time to transport the patient to SSU. We chose to use this order in all analyses, since the number of treated patients is more important, followed by the time spent in the hospital.

The number of SSU beds was the factor that affects all the chosen metrics in a significant way. Based on Lenth's method, the number of beds and seats in SSU, BA, MBA, VA, and some interactions were identified as statistically significant. The factors interaction hinders the ED performance since the locations are used together in some specific points of the model. In this way, we did not remove any variable from the analyses.

In order to obtain fit meta-models to the system and to identify eventual curvatures in the objective functions, we chose to perform a Response Surface Methodology (RSM). Using this design, we got seven central points and four axial points, resulting in another 31 experiments. A central point test the model in the upper level, lower level, and the middle of these points. We analyzed the same performance measures and it was obtained an $R^2$-adjusted fit of 90.6% for measure (i); 93.1% for (ii); 95.2% for (iii) and 98.6% for (iv) and a predicted $R^2$ of 82.0%; 72.7%; 83.5% and 95.6%, respectively. $R^2$ refers to the degree of model reliability, and higher values indicate that the model better fits its data. The value of $R^2$ is always between 0 and 100%. Therefore, the results indicated the best factors combination is:

- 30 beds in SSU;
- 45 beds in BA;
- 5 beds in MBA;
- 15 seats in VA;

The model was replicated ten times under the same conditions as the IP scenario in order to verify the optimized response. In this case, the resources do not have a set shift. Thus, analyzing the same parameters, we observed:

- On average, 1,505 patients arrived in the model;
- On average, 1,436 (95.4%) patients were completed treated in the simulated period;
- All patients make it triage;
- On average, LOS was 436.0 min;
- Patients wait about 25.7 min to be seen after triage;
- Patients who need to go to SSU wait, on average, 128.5 min to be transferred.

Compared to the IP scenario, all patients underwent triage, showing that patient flow became more continuous than in the first approach. Still, the number of treated patients increased by around 590%. On the other hand, the waiting time to be attended after triage was an average of 25.7 min, 5.5 min for CTAS I, 11.0 min for CTAS II, 19.6 min for CTAS III, 37.5 min for CTAS IV and 40.9 for CTAS V.

Despite the increasing number of patients and the decreasing in LOS and waiting time to be attended after triage, the resources were not dimensioned. In other words, the staffs are still idle. Thus, the second scenario goals to dimension the resources in each group and shift.

### Design and analysis of experiments for resources

As mentioned, in the previous scenario, the resources were not defined. We performed a DoE design $2^4$ to determine the influence of staffing on each shift for TN, EDN1, EDN2, and physicians. Furthermore, it was defined as an arrangement $2^3$ for the porters since they have only three start shift options. The maximum number of staff per hour was considered to define the "level +". Table 6 shows the level of each resource group.

The results presented by DoE show that the amount of staff in each shift is statistically significant for at least one of the performance metrics. In this way, we did not remove any input variables. After the analysis and some adjustments, it was determined the ideal number of resources in each group and shift, presented in Table 7. In addition, we proposed a change in the shift of the porters to the same as the other staff.

"Level+" was chosen mostly in the shifts from 9:00 a.m. to 9:00 p.m., because the demand is higher at these times. We decided to maintain the workload of 85% (average) for nurses and doctors because emergencies may happen. Thereby, these features can meet emergencies without compromising patients already in the ED. Defining the ideal number of resources and executing the model with ten replicates, the results show:

- On average, 1,504 patients arrived in the model;
- On average, 1,442 (95.8%) patients were completed treated in the simulated period;
- All patients make it triage;
- On average, LOS was 511.6 min;
- Patients wait about 45.8 min to be seen after triage;
- Patients who need to go to SSU wait, on average, 97.1 min to be transferred.

The workload was balanced, and the number of treated patients continues the same. However, the LOS average increased around 75 min. The waiting time to be attended after triage was, on average, 45.8 min, increasing about twice. The waiting time after screening for each CTAS was 6.6 min for CTAS I, 13.3 min for II, 24.7 min for III, 74.0 min for IV, and 138.0 for V. Finally, the last improvement proposed is related to the number and scales of the specialist in the unit care.

### Specialists in ED

When the patients need a medical appointment, they must wait until the specialist arrives at ED, because they are not available in the unit. All the specialists, except the urologist, gynecologist, otolaryngologist, and ophthalmologist, operate all the time. The other
Table 6 DoE variables levels for resources.

| | Shift | | | | Resource |
|---|---|---|---|---|---|
| | 07:00 a.m. 07:00 p.m. | 09:00 a.m. 09:00 p.m. | 11:00 a.m. 11:00 p.m. | 07:00 p.m. 07:00 a.m. | |
| Level- | 1 | 0 | 1 | 1 | TN |
| Level+ | 2 | 1 | 2 | 2 | |
| Level- | 4 | 0 | 1 | 4 | EDN1 |
| Level+ | 8 | 2 | 2 | 8 | |
| Level- | 3 | 1 | 1 | 3 | EDN2 |
| Level+ | 6 | 2 | 2 | 6 | |
| Level- | 3 | 0 | 2 | 3 | Physician |
| Level+ | 6 | 2 | 1 | 6 | |
| | 07:00 a.m. 07:00 p.m. | 10:00 a.m. 06:00 p.m. | 07:00 a.m. 07:00 p.m. | | |
| Level- | 2 | 2 | 2 | | Porter |
| Level+ | 3 | 3 | 3 | | |

Table 7 Resource number for shift and group.

| Staff | Shift | | | |
|---|---|---|---|---|
| | 07:00 a.m. 07:00 p.m. | 09:00 a.m. 09:00 p.m. | 11:00 a.m. 11:00 p.m. | 07:00 p.m. 07:00 a.m. |
| TN | 1 | 0 | 1 | 2 |
| ED1 | 4 | 2 | 0 | 3 |
| ED2 | 3 | 1 | 2 | 3 |
| Physician | 3 | 2 | 2 | 3 |
| Porter | 3 | 1 | 0 | 3 |

specialists work in shifts from 08:00 a.m. to 05:00 p.m. (on average). At other times, the internal medicine covers the cardiologist, nephrologist, and gastroenterologist. Currently, there is only one specialist available for each specialty at each shift, except the Crisis Response, which has two doctors from 08:30 a.m. to 07:30 p.m.

Moreover, it is necessary to consider the response time after calling the specialists. Psychiatrists take between 1 and 6 h to reach the unit, while neurologists can take 15 to 90 min (according to CTAS level), and Crisis Response specialists take about 5 min.

Among the 13 specialists, the most requested are psychiatrists, neurologists, and crisis response. Then, these specialists need to stay most of the time in the ED. Therefore, we made some changes in the shifts:

- Psychiatrist: a specialist all the time in the unit (avoids the delay that can reach 6 h);
- Neurologist: a specialist in the same shift of the IP scenario (08:00 a.m.–05:00 p.m.), but allocated directly to the unit (avoiding a delay that can reach 90 min);
- Crisis Response: One expert all the time on the unit rather than two on the predetermined shift (08:30 a.m.–07:30 p.m.). Although this specialist's response time is relatively short, we chose to allocate it directly to the unit.

| Table 8 | Number of locations and resources. | | | | |
|---|---|---|---|---|---|
| Local | IP | FP | Resource | IP | FP |
| BA | 45 | 45 | TN | 3 | 4 |
| MBA | 3 | 5 | EDN1 | 9 | 9 |
| SSU | 10 | 30 | EDN2 | 8 | 9 |
| VA | 10 | 15 | Physicians | 12 | 10 |
| | | | Porters | 3 | 7 |

After ten runs, the results are:

- On average, 1,496 patients arrived in the model;
- On average, 1,441 (96.3%) patients were completed treated in the simulated period;
- All patients make it triage;
- On average, LOS was 450.7 min;
- Patients wait about 19.0 min to be seen after triage;
- Patients who need to go to SSU wait, on average, 137.2 min to be transferred.

One of the actions that blocked continuous patient flow is to wait for the specialist. In this sense, the simulations show that the LOS decreases by about 11.9%. In addition, the waiting time for patients requiring a transfer to SSU is, on average, 42.8 min below that requested.

## Confirmation runs

In order to confirm the proposed changes, the model was simulated 30 times and a warm-up from Monday (00:00) to Tuesday (00:00). Data were collected from Tuesday to Tuesday. Table 8 presents the Final Planning (FP) configuration for locations and resources.

According to simulation results, the study recommends two additional beds in the MBA and five additional chairs in the VA. Regarding the resources, it is necessary to hire one TN, one EDN2, four porters, and reduce two physicians. Table 7 lists the shift for each group. It was necessary to dimension the staff, avoiding, when possible, hire them. The number of beds in SSU directly affects patients' LOS. Then, the ideal number is 30 beds, corresponding, on average, 146.4 min of waiting for transfer to SSU. About the specialist, a psychiatrist, a neurologist, and a crisis response must stay directly in the unit. With the proposed measures, the patient's LOS decreases by 1,752.6 min. Table 9 presents the results of the IP and FP scenario.

*Sargent, Goldsman & Yaacoub (2016)* suggest that, for a non-observable system, an objective approach is to compare the results from the developed model with another similar system to perform the validation. In this sense, we decide to compare the model with another ED that works in the same way to verify the results from the future state. In our model, the mean LOS is 461.2 min and a standard deviation of 20.7 min. In the other ED, the LOS is 472.1 min. Then, we carried out a 1-Sample Equivalence test that evaluates if the mean of a population is close enough to a target value, specifying a range to be equivalent. In the test, we can say that the difference between the model's results and

**Table 9  Outputs for IP and FP scenario.**

| Metrics (min) | | Initial planning | | Final planning | | Rate (mean) |
|---|---|---|---|---|---|---|
| | | Mean | Confidence interval | Mean | Confidence interval | |
| Patients | Input | 1,504 | – | 1,506 | – | – |
| | Output | 258 | – | 1,444 | – | 560% |
| | Treated (%) | 17.2 | – | 95.7 | – | 556% |
| | Triage (%) | 23.1 | – | 100.0 | – | 433% |
| | Average | 2,213.7 | (2,131.8–2,295.6) | 461.2 | (453.7–468.7) | −79% |
| | CTAS I | 2,809.9 | (1,381.3–4,750.1) | 1,072.2 | (787.6–1,356.8) | −62% |
| LOS | CTAS II | 3,077.2 | (2,778.8–3,375.6) | 918.2 | (880.9–955.5) | 70% |
| | CTAS III | 2,219.3 | (2,051.2–2,387.5) | 452.7 | (438.8–466.6) | −80% |
| | CTAS IV | 1,746.2 | (1,585.5–1,907.9) | 252.8 | (246.1–259.5) | −86% |
| | CTAS V | 1,500.8 | (903.5–2,091.0) | 299.9 | (275.4–324.4) | −80% |
| Triage to Bed | Average | 404.3 | (369.7–439.0) | 20.8 | (19.8–21.8) | −95% |
| | CTAS I | 25.9 | (4.2–47.5) | 11.2 | (6.1–17.4) | −57% |
| | CTAS II | 164.6 | (121.6–207.7) | 10.4 | (6.1–16.3) | −94% |
| | CTAS III | 332.2 | (275.2–389.3) | 13.7 | (12.9–14.5) | −96% |
| | CTAS IV | 649.7 | (559.9–739.5) | 26.9 | (25.2–28.7) | −96% |
| | CTAS V | 419.5 | (131.0–707.9) | 82.1 | (64.6–99.7) | −80% |
| | Bed to SSU | 367.4 | (253.1–481.7) | 146.4 | (133.8–159.0) | −60% |

benchmarking is not significant, less than 5% (*p*-value < 0.001). In other words, the models are equivalents, and we can affirm that the FP scenario and the results are validated.

After the proposed changes, the number of treated patients increased considerably, making their flow continuous during the process (LT principles). In addition, patients are seen as expected after triage, according to the CTAS classification level.

## CONCLUSIONS

The presented study aimed to propose an ideal state aligned with LT for a Canadian ED expansion. The purpose was to increase the number of treated patients and reduce LOS, without compromising the quality of services and patient safety. In this way, it determined the optimal number of beds for SSU, BA, MBA, and VA; defined the ideal number of resources; reduced LOS, waiting time after triage phase, and transfer to SSU. Hence, the study used the Modeling and Simulation method proposed by *Montevechi et al. (2010)*.

We built the conceptual model using the IDEF-SIM modeling technique and the input data was obtained through the ED's historical data, registrations from a system, and experts from the hospital. The computer model was built in FlexSim Healthcare® software, and validated it with experts and sensitivity analysis technique. For the experiments, at first, we carried out the DoE to verify the influence of the area expansion (BA, MBA, SSU, and VA) in the chosen metrics. We also used DoE to determine the optimal resource number at each shift and its optimal scale to meet changing demand throughout the day and week. Finally, experiments were performed to reduce patients' waiting time by specialists. Among the evaluated metrics, we chose to prioritize them as follows: (i) weekly treated patients;

(ii) LOS; (iii) time for the patient to be attended after triage and (iv) time to transport the patient to SSU.

After the analysis, the simulation indicates that the model in its IP scenario cannot meet the demand. For the FP scenario, we observed that DES and LT integrated into the DoE allowed increasing the number of patients that went through the triage process from 23.1% to 100.0%. The patient LOS reduced from 2,231.8 to 461.2 min. The resource workload was balanced according to LT principles. In this way, we got a significant improvement in the process.

Regarding the limitations of this study, we found difficulties in executing replicates in the IP scenario due to the computational effort required. For this reason, the simulation warm-up was only one day, and the simulation performed for one week (Tuesday to Tuesday). Another significant difficulty was to obtain some data from hospital processes. Even with the use of systems like Meditech®, some data are still obtained through expert estimation. According to the literature, this limitation is common in healthcare simulation projects. Finally, for future work, we suggest investigating other resources, e.g., technicians, receptionists, and ward clerk. In addition, the economic viability of different layouts can be assessed for labs and DI.

### Funding
The authors received funding from Coordenação de Aperfeiçoamento de Pessoal de Nível Superior (CAPES), the Conselho Nacional de Desenvolvimento Científico e Tecnológico (CNPq), the Fundação de Amparo à Pesquisa do Estado de Minas Gerais FAPEMIG, and Society for Health Systems. The funders had no role in study design, data collection and analysis, decision to publish, or preparation of the manuscript.

### Grant Disclosures
The following grant information was disclosed by the authors:
Coordenação de Aperfeiçoamento de Pessoal de Nível Superior (CAPES).
The Conselho Nacional de Desenvolvimento Científico e Tecnológico (CNPq).
The Fundação de Amparo à Pesquisa do Estado de Minas Gerais FAPEMIG.
Society for Health Systems.

### Competing Interests
The authors declare there are no competing interests.

### Author Contributions
- Gustavo Teodoro Gabriel conceived and designed the experiments, performed the experiments, analyzed the data, performed the computation work, authored or reviewed drafts of the paper, and approved the final draft.
- Afonso Teberga Campos conceived and designed the experiments, performed the experiments, performed the computation work, authored or reviewed drafts of the paper, and approved the final draft.

- Aline de Lima Magacho and Lucas Cavallieri Segismondi conceived and designed the experiments, performed the experiments, analyzed the data, performed the computation work, prepared figures and/or tables, and approved the final draft.
- Flávio Fraga Vilela conceived and designed the experiments, analyzed the data, performed the computation work, prepared figures and/or tables, and approved the final draft.
- José Antonio de Queiroz conceived and designed the experiments, authored or reviewed drafts of the paper, and approved the final draft.
- José Arnaldo Barra Montevechi analyzed the data, authored or reviewed drafts of the paper, and approved the final draft.

## Data Availability

The analysis of DoE, the FlexSim Healthcare software code and instructions to open the code are available in Supplemental Files.

## Supplemental Information

Supplemental information for this article can be found online at http://dx.doi.org/10.7717/peerj-cs.284#supplemental-information.

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
