# Peer review of "Lean thinking by integrating with discrete event simulation and design of experiments: an emergency department expansion"

_PeerJ Computer Science, doi:10.7717/peerj-cs.284_

## Round 0.1 · original submission · Major Revisions

Please classify the goals and motivation of the study in the Abstract, Introduction, and Conclusion.

Please present suitable comments to the related work.

Please better explain the “Design and analysis of experiments” step.

Please present more clear descriptions of the availability of the dataset.

Please give more information about DE? The name, location, etc.? If not, please explain.

Present more details in Figure 1 and Figure 2

Please improve the English language to ensure that an international audience can clearly understand your text.

Reviewer 1 ·

Basic reporting

The paper proposes to study the expansion of an emergency department. The authors propose to simulate the current emergency department to evaluate the number of resources (beds) required to meet the demands coming from close care centers and to reduce the length of the stay of the patients. The proposal combines DES and Lean Healthcare.
The conceptual model was implemented with the IDEF-SIM language and the computing modeling with Felxim.

Experimental design

Can you better explain the “Design and analysis of experiments” step. What do you mean with “we chose a complete factorial arrangement for screening the most significant decision variables”

Can you give more information about the DE? The name, location, etc.? If not, please explain.
How did you set the waiting times and the delays?

Validity of the findings

The conceptual model was validated with the review of experts. Experts verified if the model matches the real system. Figure 1, is really difficult to read. Can you better explain this figure, the text at the bottom of the figure and perhaps divide the figure? In this figure you have 3 flows, can you detail which process correspond to each flow?
In section “Model Execution and Initial Analysis” how many days did you simulate? Which is the inter-arrival rate of patients? Did they arrive all at the same time?

Figure 2 is not explained. Why you use an alpha =0.5? Which is the effect of alpha? What does it means an standardized effect of 200 or 20?


In line 318, what are central points and axial points?
What is “R2 fit”?
In line 322 you say “The results indicated that the best combination of factors is:” How did you get these results?
In line 357: “After the analysis and some adjustments, we determined the ideal number of resources” How did you get this ideal number?

Additional comments

In the introduction autos claim that DES has no additional costs. However, it depends on how you define costs. In fact, there is a cost in terms of time, model designs, adjusting the simulation parameters and validation.
“we proposed improvements by the principles of patient continuous Flow”, which ones?
In line 276, what do you mean with: “increasing the amount of time to add value”


In general, this is an interesting work. However, there are several issues not properly addressed in this paper. The inter-arrival speed of patients is not defined. How do you decide to increase the number of resources is arbitrary. I would expect a more methodologic experiment design to decide how to increase the number of resources, etc. And more importantly, it is not clear how you apply Lean healthcare in this work. Additionally, it would be nice to see the Felxim model used for the experiments.

The English language should be improved to ensure that an international audience can clearly understand your text.
Some examples:
the literature defines as the model development process
allowing observing if there is an iteration between  iteration or interaction?
aims to analyses its expansion
and VA, and some interactions-> Remove “and”

Reviewer 2 ·

Basic reporting

(1)
The study refers to the hospital under consideration in an inexplicably vague way, as ‘Canadian emergency department’.

The details of the hospital (e.g. name, precise location etc.) should be clearly mentioned.
If there are issues to prevent mentioning such info, then compelling reasons should be given.

(2)
The goals and motivation of the study are mentioned in a very broad perspective, rather than well-defined.The authors need to be more specific about the key motivation behind the study. The reader would be wondering was this work initiated by a request from the hospital, related stakeholders, or the authors themselves?

Stating that the goal was to plan the expansion of a Canadian ED is not quite enough. Simulation studies should include clear-cut ‘questions’ of interest. The care process under consideration should be mentioned clearly, as they would not be modeling the whole processes involved. I'd recommend adding a section as 'Motivation' or 'Questions of Interest'.

(3)
The study gave a good review of literature and good examples of relevant work.
However, the review unfortunately does not give any critique or positioning of this study with respect to that literature.

The authors only mentioned (briefly) that they contribute to the literature by using Design of Experiments. I’m afraid it is not clear at all how this would contribute to literature.

(4)
Data description is an important missing piece. A thorough description of the fields included in the medical/process records should be provided. A snapshot of a few records may be presented for the purpose of demonstration. The authors may also refer to a data dictionary if available.

(5)
How the authors actually acquired the data?
Did they use a freely accessible dataset?
Did they submit some request to access the data?
Did they need an approval of research ethics ?

(6)
It’s good to see that the authors are aware of the study limitations. However, I am still wondering whether there could be (under-estimated) limitations pertaining to the quality of data used. Based on my experience with healthcare data, this has been always a concern. This is something that should be properly considered in the limitations.

(7)
I am afraid that the quality of language should be reviewed thoroughly. Many grammar and language mistakes plainly exist.

Experimental design

(1)
Most importantly, the study did not mention any convincing reasons behind using the DES approach. There are actually three approaches of simulation modeling: i) System Dynamics , ii) DES , and iii) Agent-based models.

Going with one of those approaches depends largely on the modeller’s view of the world (i.e. the problem under study). From the study perspective, were patients viewed as aggregates sharing similar characteristics (i.e. System Dynamics), or individual entities with particular characteristics (i.e. DES), or agents with particular behaviour (i.e. Agent-Based)? This needs to be clearly explained.

Sally Brailsford provides a very useful comparison on that issue in the study below. Please consider looking at it, especially Section 8.
Brailsford, S.C. and Hilton, N.A., 2001. A comparison of discrete event simulation and system dynamics for modelling health care systems.

(2)
Please describe in detail the entities involved in the simulation model.
In addition to Figure 1, a further explanation of the timing/events in the simulation model would be helpful for the reader.

Validity of the findings

(1)
As a simulation study, I'd have expected more specific details regarding the verification tests used to verify the simulation model. Examples may be: i) Structure-verification test, ii) Extreme conditions test, or iii) Parameter-verification Test.

(2)
It is hard to find a clear interpretation of the simulation output. The study largely reported the simulation runs and parameters, but the interpretation is ill-described I'm afraid. The discussion of results should be linked to the questions of interest (which are already missing).

---

## Round 0.2 · Minor Revisions

The authors are suggested to further revise the manuscript according to the reviewer’s valuable comments, by specifically focusing on the following issues:

(1) improve the abstract to make it more informative;

(2) revise the literature review;

(3) also clearly describe the data description;

(4) revise the writing language through the text;

(5) re-draw Figure 1 and Figure 3. Figure 1 is large, but not clear enough. Figure 3 is also not high-quality. Please use Origin and Illustrator to draw high-quality vector figures.

Reviewer 2 ·

Basic reporting

The article has largely improved, and I thank the authors for considering the earlier comments thoroughly. There are some issues that still need to be covered as follows.

(1)
The abstract should be more informative by including more details about the case study under consideration. The reader should get more information about the use case, and the key findings of the study.

(2)
For better readability, the literature review could be divided into smaller sections.

Further, the review can include a background section to get the reader familiar with the main tools used in the study. For example, the IDEF-SIM language can be described in that section. This would avoid explaining such issues in the experimental part.
Also, I recommend mentioning some representative examples in literature that successfully applied the IDEF-SIM in similar contexts.

(3)
Data Description should have a separate section. A better presentation is needed to fully describe the dataset acquired.
Summary statistics of the dataset and its fields should be useful.
Figures and tables can be used effectively in this regard.

(4)
The quality of writing has been revised to an adequate level, though there are still several language mistakes.
The abstract particularly needs further revision.

Experimental design

A sufficient explanation of the simulation entities and their characteristics is still largely missing. For example, what are the particular characteristics of patients included in the simulation model.
Were those characteristics learned from the dataset or some domain knowledge ?

Validity of the findings

The major weak point of the manuscript is the lack of a convincing validation.
The validation is a core component of simulation studies. Without a clear-cut validation, the simulation output would be considered with a large degree of skepticism.

The authors already cite an excellent reference in the context of simulation validation, which is (Sargent, 2013). Please apply any of the validation methods mentioned in Section 3.2 at (Sargent, 2013).

---

## Round 0.3 · Minor Revisions

I am happy that the manuscript has been largely improved. And, please consider and revise the points suggested by the reviewer in the final version.

Reviewer 2 ·

Basic reporting

I’d like to thank the authors for working thoroughly on the previous comments.
The manuscript has been largely improved.
Please consider the points below in the final version.

(1)
It is not entirely clear if the simulation model is simulating the current situation of ED (under full demand), or it predicts a possible scenario in the future after closing the surrounding care centres.
I am wondering if those centres have been already closed?
If yes, would this decision be possibly related to some economic feasibility?
This generally raises key questions about the purpose of the study.

(2)
A short introductory paragraph would be encouraged at the beginning of the Background section.
This may help prepare the reader about the topics reviewed.

(3)
The language has improved, though there are still several mistakes.
For example:
line 324: Patients (of) CTAS level III and IV
line 333: needs (an) exam
line 344: Table 4 describes (these) data.

Also, please use either British or American English consistently.
For example, analyse (British) / Analyze (American)

Experimental design

It seems unrealistic to assume that staff meetings can be completely removed.
Meetings might be reduced to a sufficient level, but it would not be reasonable to assume that the medical staff need no meetings at all.
There are activities that do add value to the patient’s journey, though indirectly.

Validity of the findings

I am concerned with the simulation output. I believe that the simulated scenarios need to strike a balance regarding the improvement before and after applying the proposed procedures.
For example, the percentage of treated patients went from 17% to over 95%, this seems overly optimistic.
The reader would be left wondering how such low percentages could happen in Canada, reportedly as one of the most performing healthcare systems in the world.

---

## Round 0.4 · accepted · Accept

I am happy that the authors further improve their manuscript according to the second round of reviewing comments. I think now it is ready for publication.